# Peukert’s Law-Based State-of-Charge Estimation for Primary Battery Powered Sensor Nodes

**DOI:** 10.3390/s23021036

**Published:** 2023-01-16

**Authors:** Hongli Dai, Yu Xia, Jing Mao, Cheng Xu, Wei Liu, Shunren Hu

**Affiliations:** 1School of Electrical and Electronic Engineering, Chongqing University of Technology, Chongqing 400054, China; 2College of Electronic and Information Engineering, Nanjing University of Aeronautics and Astronautics, Nanjing 210016, China

**Keywords:** state-of-charge (SOC), estimation, Peukert’s Law, primary battery, sensor node, wireless sensor networks (WSNs)

## Abstract

Accurate state-of-charge (SOC) estimation is essential for maximizing the lifetime of battery-powered wireless sensor networks (WSNs). Lightweight estimation methods are widely used in WSNs due to their low measurement and computation requirements. However, accuracy of existing lightweight methods is not high, and their adaptability to different batteries and working conditions is relatively poor. This paper proposes a lightweight SOC estimation method, which applies Peukert’s Law to estimate the effective capacity of the battery and then calculates the SOC by subtracting the cumulative current consumption from the estimated capacity. In order to evaluate the proposed method comprehensively, different primary batteries and working conditions (constant current, constant resistance, and emulated duty-cycle loads) are employed. Experimental results show that the proposed method is superior to existing methods for different batteries and working conditions, which mainly benefits from the ability of Peukert’s Law to better model the rate-capacity effect of the batteries.

## 1. Introduction

With the rapid development of sensors, embedded computing, and wireless communication technology, wireless sensor networks (WSNs) have been widely used in many applications [1,2]. As it is difficult, if not impossible, to provide wired power supply in many scenarios, such as areas that are inaccessible or dangerous to humans, sensor nodes are usually powered by batteries [3]. From another point of view, using a battery instead of a wired power supply makes node deployment more flexible, which helps to achieve the goal of self-organization.

Both a primary and secondary battery have been widely used in WSNs. Sensor nodes powered by a secondary battery (also called a rechargeable battery) could replenish electricity through environmental energy harvesting technologies [4,5]. However, not all scenarios can effectively harvest energy from the environment, such as the nodes deployed under roads for vehicle detection [6]. In addition, for applications that are discarded after use, using secondary battery will make the cost much higher. Therefore, primary batteries such as alkaline batteries and carbon–zinc batteries are still an important power supply for WSNs [6,7,8,9,10].

The limited capacity of the primary battery determines the lifetime of sensor nodes and networks. It is necessary to dynamically schedule node tasks and network resources according to the state of the battery to utilize its capacity effectively. Therefore, estimating the state of the battery accurately, especially the state-of-charge (SOC), is essential for maximizing the lifetime of nodes and networks [11,12,13,14]. Among existing SOC estimation methods, the ones based on electrochemical, analytical, or stochastic models are more accurate [15,16,17,18,19]. However, computation complexity and measurement difficulty of these methods make them unsuitable for sensor nodes.

Therefore, lightweight SOC estimation methods specially tailored for WSNs are more popular. Among them, the fitting models based on terminal voltage [20,21] and cumulative models based on current consumption [22,23,24] are widely used in WSNs because they are much simpler. Unfortunately, accuracy of these methods is not high, and their adaptability to different batteries and working conditions is relatively poor. With this in mind, this paper firstly analyzes theoretically the estimation error of the cumulative models based on current consumption and then proposes a lightweight SOC estimation method, which applies Peukert’s Law to estimate the effective capacity of the battery and then calculates the SOC by subtracting the cumulative current consumption from the estimated capacity. To verify the accuracy of the proposed method, different primary batteries are employed and different working conditions are designed. Experimental results show that compared with existing methods, the proposed method is more accurate for different batteries and working conditions.

### 1.1. Main Contributions of This Paper

The major contributions are summarized as follows: (1) Applicability of Peukert’s Law for commonly used primary batteries is verified. (2) Estimation error of the cumulative models based on current consumption is analyzed theoretically and verified with experiments. (3) A lightweight and accurate SOC estimation method based on Peukert’s Law is proposed, which considers the rate-capacity effect of the battery more reasonably. (4) The proposed method is verified and compared with existing ones using different primary batteries and working conditions (constant current, constant resistance, and emulated duty-cycle loads).

### 1.2. Organization of This Paper

The rest of this paper is organized as follows. In Section 2, related work is reviewed. Existing lightweight SOC estimation methods are described in detail in Section 3, and the proposed estimation method is explained in Section 4. This is followed by the experimental setup in Section 5. Experimental results are discussed in Section 6. Finally, conclusions are presented and suggestions are made for future work.

## 2. Related Work

This section describes the previous work related to this paper. Firstly, by analyzing the limitation of battery capacity in WSNs, the importance of SOC estimation is presented. Then, popular SOC estimation methods for WSNs are summarized, especially the lightweight methods studied in this paper. Finally, the work related to Peukert’s Law is summarized, as Peukert’s Law is the basis of the estimation method proposed in this paper.

### 2.1. Power Supply for Sensor Nodes

Both a primary and secondary battery have been widely used in WSNs. In some outdoor scenarios, sensor nodes powered by a secondary battery could replenish electricity through solar energy harvesting [4,5]. However, not all scenarios can effectively harvest energy from the surrounding environment. In these cases, primary batteries are more appropriate [6,7,8,9,10]. For example, Bajwaa et al. proposed a wireless sensor network that estimates the weight of moving vehicles from pavement vibrations caused by vehicular motion [6]. In their work, sensor nodes are powered by a primary battery (specifically, a lithium–thionyl chloride battery), because they are buried under the road and cannot harvest energy from the environment. However, regardless whether a primary battery or secondary battery is used, the capacity is limited, as energy harvesting technologies are greatly affected by the environment and usually difficult to provide unlimited energy at any time.

### 2.2. Battery-Aware Power Management

As battery capacity is very limited, an important optimization goal of WSNs is power minimization. Many dynamic power management (DPM) mechanisms have been invented to minimize the power consumption of sensor nodes. However, it is generally difficult to utilize battery capacity efficiently if the state of the battery is not considered in the DPM. It makes battery-aware or battery-driven power management more important for WSNs. Recent studies have fully shown that considering the state of the battery when designing medium-access control protocols, trajectory-tracking algorithms, and data-transmission policies will help to prolong the lifetime effectively [12,13,14]. Therefore, accurate estimation methods for the state of the battery, especially those for SOC, are necessary for maximizing the lifetime of sensor nodes and networks.

### 2.3. SOC Estimation Methods for WSNs

Existing methods for SOC estimation include models based on electrochemical, analytical, and stochastic techniques, among others. For example, the KiBaM model, based on the chemical kinetic process of the battery, has been widely used in UAV, smart phones, and other fields [15,16]. Additionally, some complex algorithms such Bayesian inference [17,18] and Kalman filter [19] have also been used in SOC estimation. Although these models are more accurate, their computation complexity and measurement difficulty are too high for resource-constrained sensor nodes. Therefore, this paper only considers lightweight SOC estimation methods especially tailored for WSNs.

Considering that the battery terminal voltage is directly proportional to SOC, estimation models can be established based on terminal voltage measurement. The linear voltage model (LVM) is a common SOC estimation method used in the early stage [22,23]. LVM assumes that the battery is a linear voltage source. Although it is the simplest, its accuracy is difficult to guarantee. In order to improve the accuracy, Valle et al. firstly acquired the residual capacity and terminal voltage curve using MicaZ nodes powered by a carbon–zinc battery and then established the SOC estimation model through polynomial fitting [20]. However, as the residual capacity and terminal voltage curve is greatly affected by the load, accuracy of this method is still not high.

In addition to the fitting models based on terminal voltage, the cumulative models based on current consumption (also known as ampere hour integration methods) are also common SOC estimation methods used in the early stage [21,24,25]. These methods require the sensor node to provide data about the working currents. At present, several cost-efficient circuit implementations have been proposed to add current measurement capability to sensor nodes [21,26,27]. Traditional ampere hour integration methods usually use the nominal capacity of the battery as its initial capacity. However, the nonlinearity of batteries makes the actual available capacity not equal to the nominal one, which degrades the accuracy of these methods inevitably.

To solve this problem, Cunha et al. proposed an effective discharge-rate-based model (EDRM), which accounts for the effect of discharge rate on the effective capacity [21,24]. The effective capacity is modeled as a polynomial function of the current consumed by the sensor nodes. Experimental results show that EDRM is more accurate than LVM and traditional ampere hour integration methods. Rukpakavong et al. suggested that Peukert’s Law be applied to correct the cumulative current consumption, and proposed the dynamic node lifetime estimation (DNLE) method [28]. The authors claimed that DNLE is more accurate than traditional ampere hour integration methods.

### 2.4. Peukert’s Law

Peukert’s Law was proposed to describe the discharge characteristics of lead–acid batteries [29]. Owing to its low complexity and simple modeling process, Peukert’s Law has also been applied to lithium-ion batteries, supercapacitors, nickel–cadmium batteries, etc., in recent years [30,31,32,33,34,35,36]. However, it is mainly used for discharge time prediction with the known current profile, rather than online estimation of SOC [30,31,32,35,36]. Yang et al. studied the dependence of the Peukert constant of supercapacitors on voltage, aging, and temperature, and pointed out that the Peukert constant increases when the temperature is lower, although the change is moderate [32]. Xie et al. studied the impact of battery thermal evolution on the Peukert constant of LiFePO_4_ batteries and also found that the Peukert constant is little affected by temperature [33].

## 3. Problem Definition and Design Motivation

Existing studies have shown that accuracy of the cumulative models based on current consumption is significantly higher than that of the fitting models based on terminal voltage [21,24,28]. This paper intends to further improve the accuracy of the cumulative models based on current consumption. This section explains the design motivation behind the proposed method by theoretically analyzing the estimation error of the cumulative models based on current consumption.

### 3.1. Typical SOC Estimation Methods for WSNs

Firstly, typical SOC estimation methods used in WSNs are described in detail, and their mathematical expressions are given. Only lightweight estimation methods are considered because they are more suitable for resource-constrained sensor nodes. The methods listed in this section are also used to compare with the method proposed in this paper to verify its accuracy. For ease of description, Table 1 summarizes the variables used in subsequent analysis.

#### 3.1.1. Models Based on Terminal Voltage 

LVM is a common SOC estimation method used in the early stage [20,21]. The modeling process is as follows: Firstly, the terminal voltage and current are recorded when discharging the battery with a specific load. Then, SOC is calculated and the linear model of SOC and terminal voltage is fitted, as shown in (1):(1)$$SOC_{est}=a_{1}V_{t}+a_{0}$$

The modeling process for the polynomial voltage model (PVM) is similar to that of LVM. Firstly, the terminal voltage and current are recorded when discharging the battery with a specific load. Then, SOC is calculated and the polynomial model of SOC and terminal voltage is fitted, as shown in (2). Existing studies have shown that a good fitting effect could be obtained by using a third-order polynomial [25].
(2)$$SOC_{est}=a_{n}V_{t}^{n}+a_{n-1}V_{t}^{n-1}+\cdots +a_{1}V_{t}+a_{0}$$

#### 3.1.2. Models Based on Current Consumption 

EDRM accounts for the effect of discharge rate on the effective capacity, which is modeled as a function of the current consumed by the sensor node [22,23], as shown in (3). After the effective capacities under different discharge rates are tested, the function between them can be modeled by polynomial fitting.
(3)$$SOC_{est}=(1-\frac{\sum_{i=0}^{i=n}I_{i}\Delta t}{C_{est}(I_{n})})\times 100\%$$

DNLE applies Peukert’s Law to correct the cumulative current consumption to introduce the rate-capacity effect [28], as shown in (4). The nominal capacity of the battery is used in DNLE. The Peukert constant *k* should be determined for the operation of DNLE, which can be obtained by using the discharge time under different discharge rates. Peukert’s Law is also used in the method proposed in this paper. Different from DNLE, Peukert’s Law is applied to estimate the effective capacity of the battery instead of correcting the cumulative current consumption. Experimental results show that the approach adopted in this paper is more accurate and can be adapted to different batteries and working conditions. See Section 4 for detailed analysis.
(4)$$SOC_{est}=(1-\frac{\sum_{i=0}^{i=n}I_{i}^{k}\Delta t}{C_{nominal}})\times 100\%$$

### 3.2. Error Analysis and Design Motivation

SOC estimation error of the cumulative methods based on current consumption can be expressed as shown in (5). It can be seen that the estimation error of such methods presents several obvious characteristics: First, the estimation error in the initial stage is very low, because the accumulated current consumption is very small compared with the actual capacity. Second, with the increase in accumulated current consumption, the estimation error continues to increase, because $\sum_{i=0}^{i=n}I_{i}\Delta t/C_{real}$ continues to increase. Third, the maximum error depends on the accuracy of the estimated capacity *C_est_*. The closer *C_est_* is to the actual capacity *C_real_*, the smaller the error is, and vice versa.
(5)$$\begin{array}{l} Error=SOC_{est}-SOC_{real}\\ \;\;\;\;\;\;\;\;\;\;=(1-\frac{\sum_{i=0}^{i=n}I_{i}\Delta t}{C_{est}})-(1-\frac{\sum_{i=0}^{i=n}I_{i}\Delta t}{C_{real}})\\ \;\;\;\;\;\;\;\;\;\;=\sum_{i=0}^{i=n}I_{i}\Delta t(\frac{1}{C_{real}}-\frac{1}{C_{est}})\\ \;\;\;\;\;\;\;\;\;\;=\frac{\sum_{i=0}^{i=n}I_{i}\Delta t}{C_{real}}(\frac{C_{est}-C_{real}}{C_{est}}) \end{array}$$

Therefore, SOC estimation error can only be reduced effectively by improving the estimation accuracy of the effective capacity of the battery. Obviously, accuracy of the traditional ampere hour integration methods is bound to be very low because it uses the fixed nominal capacity as the effective capacity. In order to improve the estimation accuracy of the effective capacity, nonlinear characteristics of the battery must be considered, the most important of which is the rate-capacity effect. However, existing methods did not solve this problem very well. Considering the advantage of Peukert’s Law in modeling the rate-capacity effect, introducing it into the estimation of effective capacity will certainly improve the accuracy of SOC estimation.

## 4. SOC Estimation Based on Peukert’s Law 

### 4.1. Peukert’s Law

Peukert’s Law models the relationship of discharge time and discharge rate, as shown in (6):(6)$$Q=I^{k}t$$
where *Q* is an empirical constant. Peukert constant *k* is related to the material, structure, and other battery parameters. In order to obtain the value of *k* and *Q*, taking the logarithm of (6) and rearranging gives:(7)lnt=−klnI+lnQ
which means that the relationship between ln *I* and ln *t* is linear. Therefore, after discharging the battery with different *I* and obtaining corresponding discharge time *t*, *k* and *Q* can be calculated by using linear regression.

### 4.2. SOC Estimation Method Based on Peukert’s Law 

This paper applies Peukert’s Law to estimate *C_est_* and proposes a novel SOC estimation method. Although existing studies have verified the applicability of Peukert’s Law for lithium-ion batteries and supercapacitors, among others [29,30,31], its applicability for the commonly used primary batteries remains unrealized. Therefore, the applicability of Peukert’s Law for the commonly used primary batteries was verified initially and the results are analyzed in Section 6.1.

Firstly, Peukert’s Law is processed to reflect the relationship between discharge capacity and discharge rate. Similar to the original Peukert’s Law, it is still treated under the assumption of constant discharge rate. When the battery is discharged with constant rates, its capacity is the product of discharge rate and discharge time, that is:(8)$$C_{est}=It$$
where *C_est_* denotes the estimated effective capacity. Combined with Peukert’s Law, there is:(9)$$Q=C_{est}I^{k-1}$$

Further, the relationship between *C_est_*, *k* and *Q* can be obtained:(10)$$C_{est}=\frac{Q}{I^{k-1}}$$

Using this relationship, a Peukert’s Law-based SOC estimation method (PLM) is proposed, expressed as follows:(11)$$SOC_{est}=(1-\frac{(\sum_{i=0}^{i=n}I_{i}\Delta t)\times I_{n}^{k-1}}{Q})\times 100\%$$

## 5. Experimental Setup

This section describes the experimental setup, including the testbed for battery discharging, the primary batteries used, and the working conditions designed to obtain data for modeling and verification. On this basis, the fitted models for all the candidate estimation methods used in this paper are given.

### 5.1. Testbed for Discharging

In this work, the CT3001C microcurrent battery test system from LAND Electronic (http://www.whland.com/en (accessed on 15 October 2022)) was used to discharge the battery, as shown in Figure 1. The reasons for choosing a special battery test system instead of actual sensor nodes are twofold: First, the CT3001C not only supports common constant current and constant resistance discharge tests, but also supports discharge tests with programmable loads, which could be used to emulate the actual loads of sensor nodes. Second, the CT3001C has eight independent programmable channels, which can support eight-channel parallel discharge tests. As the working current of sensor nodes is usually below 50 mA, the time of a single discharge test is typically as high as several hundred hours. Therefore, a multichannel parallel test can effectively shorten the whole time required for all the discharge tests.

### 5.2. Target Batteries

Commonly used primary batteries include alkaline batteries, carbon–zinc batteries, coin-cell batteries, and others. Additionally, the commonly used WSN nodes, such as MicaZ [37] and TelosB [38], all use AA-size batteries. Therefore, the coin-cell battery was not considered in this paper and only AA-size alkaline and carbon–zinc batteries were employed for testing. The specific models are the LR6AA alkaline battery from NanFu Battery (https://www.nanfu.com/products.html (accessed on 30 December 2022)) and the R6PNU carbon–zinc battery from Panasonic (https://consumer.panasonic.cn (accessed on 20 December 2022)). Considering that the cut-off voltage for typical sensor nodes is usually not lower than 1.8 V, and two AA batteries are required in series, the cut-off voltage for discharge tests was set to 0.9 V. 

### 5.3. Working Conditions

Three different working conditions were used in this experiment, namely constant current, constant resistance, and emulated duty-cycle loads, to fully evaluate the accuracy of the proposed method. Similar to typical practices [20,21,22,23,28], constant current discharge tests were used for modeling. Considering the actual range of working currents for sensor nodes [39], 10, 20, 30, 40, and 50 mA were used. In order to ensure reliability, the discharge test for each current was carried out twice. For model verification, all three working conditions were used. Likewise, a discharge test under each configuration was also carried out twice. Configurations for verification tests are described as follows:(1)Verification with constant resistance discharge

For constant resistance discharge, 50 and 70 Ω were used. The selection of these resistances makes the current in the range 10 to 50 mA in the whole discharge process. As the current changes continuously and slowly during constant resistance discharge, its effect on the accuracy of the SOC estimation methods can be evaluated.

(2)Verification with emulated constant current discharge

Typical working currents of MicaZ nodes were used for constant current discharge verification. Two situations were emulated: First, only the MCU of MicaZ works, for which the typical current is 8 mA [39]. Second, the MicaZ is in receive-state, for which the typical current is 23.3 mA [39].

(3)Verification with emulated duty-cycle loads

In practical applications, sensor nodes usually work in duty-cycle modes to minimize power consumption [12]. Therefore, emulating the actual node loads helps to analyze the accuracy of the SOC estimation methods more realistically. Duty cycles of 50%, 20%, 10%, and 5% were emulated. In particular, discharging 100 ms with 8 mA and then discharging 100 ms with 23.3 mA emulates a duty-cycled discharge with a 50% duty cycle. Similarly, discharging 400, 900, and 1900 ms with 8 mA and then discharging 100 ms with 23.3 mA emulates duty-cycled discharges with 20%, 10%, and 5% duty cycles, respectively.

### 5.4. Fitted Estimation Methods

As the time of a single discharge test is as high as several hundred hours and there are many combinations of battery types and working conditions, even though eight-channel parallel test was adopted, the whole experiment still lasted nearly 2 months. Figure 2 shows the discharge characteristics of alkaline and carbon–zinc batteries under constant current discharge tests. All SOC estimation methods were fitted using the same data, and the results are shown in Table 2 and Table 3. Among them, LVM and PVM were fitted with the mean value of data under different discharge rates. It should be pointed out that the model parameters for these SOC estimation methods are inevitably affected by the discharge conditions, such as the cut-off terminal voltage and the experimental temperature. If the discharge conditions change, model parameters of these SOC estimation methods should be fitted again using the discharge data of the corresponding conditions.

## 6. Performance Verification and Analysis

### 6.1. Verification of Peukert’s Law

Firstly, the applicability of Peukert’s Law for the commonly used primary batteries was verified. The relationship between discharge rate and discharge time for the alkaline battery is shown in Figure 3a. It can be seen that ln *I* and ln *t* have an obvious linear relationship. It means that Peukert’s Law is applicable within the target range of currents for the alkaline battery. The fitted Peukert constant *k* = 1.06 and *Q* = 3651.89. Similarly, Peukert’s Law is applicable for the carbon–zinc battery, as shown in Figure 3b, for which the fitted Peukert constant *k* = 1.07 and *Q* = 1245.84. Existing studies have pointed out that parameters *k* and *Q* are not fixed and affected by several factors, the important one of which is the cut-off voltage. Therefore, *k* and *Q* were fitted using different cut-off voltages, as shown in Figure 4. It is obvious that both *k* and *Q* change with the cut-off voltage. Therefore, it is necessary to determine the value of *k* and *Q* according to the target cut-off voltage in practice.

### 6.2. Accuracy of SOC Estimation for the Alkaline Battery

This section compares the accuracy of all the candidate SOC estimation methods for the alkaline battery by using the discharge data for verification, as described in Section 5.3. Firstly, discharge characteristics under different working conditions are presented. Then, the accuracy of all the candidate SOC estimation methods under these working conditions is analyzed.

#### 6.2.1. Discharge Characteristics under Different Working Conditions

Figure 5a shows the terminal voltage of the battery during the whole discharge process. It is obvious that the curves under constant current and constant resistance discharge are much smoother than those under duty-cycle discharge. From the enlarged part of the curves in Figure 5a, it can be seen that the terminal voltage of the battery is not monotonically decreasing under duty-cycle discharge. Instead, fluctuating voltage appears. This is because the recovery effect works under duty-cycle discharge, which makes the terminal voltage of the battery increase when the discharge rate becomes smaller. Figure 5b shows the relationship of SOC and terminal voltage for the battery under different working conditions. It is obvious that fluctuation of the terminal voltage under duty-cycle discharge makes SOC and terminal voltage no longer in one-to-one correspondence. Undoubtedly, this leads to poor performance of the fitting models based on terminal voltage.

#### 6.2.2. Verification with Constant Current Discharge

Figure 6a shows the actual SOC and estimated SOC of all the candidate methods under constant current discharge. Obviously, errors of the fitting models based on terminal voltage are much larger. Additionally, they behave very differently under different discharge rates. Owing to the difference in the initial voltage of the batteries, the estimated SOC of the fitting models based on terminal voltage even exceeds 100% in the initial stage. Compared with the fitting models based on terminal voltage, accuracy of the cumulative models based on current consumption is much higher, especially for EDRM and PLM proposed in this paper.

In order to describe the performance of the cumulative models based on current consumption more clearly, Figure 6b shows the average error for every 10% SOC interval. It can be seen that the error variation in such methods conforms to the theoretical analysis in Section 3.2 very well. That is, the estimation error in the initial stage is very small, and it continues to increase with the increase in accumulated current consumption. The estimation error of DNLE increases very fast as it applies Peukert’s Law to correct the cumulative current consumption. In contrast, much better results are obtained when PLM applies Peukert’s Law to estimate the effective capacity of the battery. The accuracy of EDRM is close to but a little less than that of PLM. More importantly, PLM proposed in this paper is more adaptive to different discharge currents.

#### 6.2.3. Verification with Constant Resistance Discharge

Figure 7a shows the actual SOC and estimated SOC of all the candidate methods under constant resistance discharge. Performance of these methods is basically consistent with that under constant current discharge. It can be seen that errors of the fitting models based on terminal voltage are still much larger and they behave very differently under different resistances. Compared with the fitting models based on terminal voltage, accuracy of the cumulative models based on current consumption is much higher, especially for EDRM and PLM proposed in this paper. Figure 7b shows the average error for every 10% SOC interval. Obviously, the error variation in such methods conforms to the theoretical analysis in Section 3.2 very well. The estimation error of DNLE still increases very fast. The accuracy of EDRM is close to but a little less than that of PLM. More importantly, PLM proposed in this paper is more adaptive to different discharge resistances.

#### 6.2.4. Verification with Emulated Duty-Cycle Discharge

Figure 8a shows the actual SOC and estimated SOC for all the candidate methods under emulated duty-cycle discharge. The fitting models based on terminal voltage still perform poorly. Compared with the fitting models based on terminal voltage, accuracy of the cumulative models based on current consumption is much higher, especially EDRM and PLM proposed in this paper. Figure 8b shows the average error for every 10% SOC interval. The estimation error of DNLE increases much faster under duty-cycle discharge. Obviously, it can be seen that the accuracy of both PLM and EDRM becomes worse under duty-cycle discharge. This is because both of these models only consider the rate-capacity effect of the battery. When discharging in duty-cycle modes, other nonlinear effects of the battery such as recovery effect come into play and make the performance of PLM and EDRM worse. This can also be confirmed by the discharge characteristics shown in Figure 5. However, compared with EDRM, PLM has higher accuracy. Table 4 summarizes the average error of all the candidate methods for the alkaline battery under different working conditions. It is obvious that PLM is the best under all working conditions owing to Peukert’s Law’s ability to better model the rate-capacity effect.

### 6.3. Accuracy of SOC Estimation for the Carbon–Zinc Battery

This section compares the accuracy for all the candidate SOC estimation methods for the carbon–zinc battery, using the discharge data for verification, as described Section 5.3. Firstly, discharge characteristics under different working conditions are presented. Then, the accuracy of all the candidate SOC estimation methods under these working conditions is analyzed.

#### 6.3.1. Discharge Characteristics under Different Working Conditions

The discharge characteristics of the carbon–zinc battery are basically consistent with those of the alkaline battery. Figure 9a shows the terminal voltage of the battery during the whole discharge process. It is obvious that the terminal voltage of the battery is also not monotonically decreasing under duty-cycle discharge. Instead, fluctuating voltage appears. This is because the recovery effect works under duty-cycle discharge, which makes the terminal voltage of the battery increase when the discharge rate becomes smaller. Figure 9b shows the relationship of SOC and terminal voltage of the battery under different working conditions. Similarly, SOC and terminal voltage are also no longer in one-to-one correspondence under duty-cycle discharge. Undoubtedly, this leads to poor performance of the fitting models based on terminal voltage.

#### 6.3.2. Verification with Constant Current Discharge

The results and conclusions for the carbon–zinc battery are basically consistent with those for the alkaline battery. Figure 10a shows the actual SOC and estimated SOC for all the candidate methods under constant current discharge. It can be seen that errors of the fitting models based on terminal voltage are still much larger and they behave very differently under different discharge rates. Owing to the difference in the initial voltage of the batteries, the estimated SOC of the fitting models based on terminal voltage even exceeds 100% in the initial stage. Compared with the fitting models based on terminal voltage, accuracy of the cumulative models based on current consumption is much higher, especially for EDRM and PLM proposed in this paper. Figure 10b shows the average error for every 10% SOC interval. Obviously, error variation in such methods conforms to the theoretical analysis in Section 3.2 very well. It can be seen that the estimation error of DNLE still increases very fast. The accuracy of EDRM is close to but a little worse than that of PLM. More importantly, PLM proposed in this paper is more adaptive to different discharge rates.

#### 6.3.3. Verification with Constant Resistance Discharge

Figure 11a shows the actual SOC and estimated SOC for all the candidate methods under constant resistance discharge. It can be seen that errors of the fitting models based on terminal voltage are still much larger and they behave very differently under different discharge resistances. Owing to the difference in the initial voltage of the batteries, the estimated SOC of the fitting models based on terminal voltage even exceeds 100% in the initial stage. Compared with the fitting models based on terminal voltage, accuracy of the cumulative models based on current consumption is much higher, especially for EDRM and PLM proposed in this paper. Figure 11b shows the average error for every 10% SOC interval. It can be seen that error variation in such methods conforms to the theoretical analysis in Section 3.2 very well. Obviously, the accuracy of EDRM is close to but a little worse than that of PLM. More importantly, PLM proposed in this paper is more adaptive to different resistances.

#### 6.3.4. Verification with Emulated Duty-Cycle Discharge

Figure 12a shows the actual SOC and estimated SOC of all the candidate methods under emulated duty-cycle discharge. The fitting models based on terminal voltage still perform poorly. Compared with the fitting models based on terminal voltage, accuracy of the cumulative models based on current consumption is much higher, especially for EDRM and PLM proposed in this paper. Figure 12b shows the average error for every 10% SOC interval. Obviously, error variation in such methods conforms to the theoretical analysis in Section 3.2 very well. The estimation error of DNLE increases much faster under duty-cycle discharge. Obviously, it can be seen that the accuracy of both PLM and EDRM becomes worse under duty-cycle discharge. This is because both these models only consider the rate-capacity effect. When discharging in duty-cycle modes, other nonlinear effects of the battery such as recovery effect come into play and make the performance of PLM and EDRM worse. This can also be confirmed by the discharge characteristics shown in Figure 11. However, compared with EDRM, PLM has higher accuracy. Table 5 summarizes the average error of all the candidate methods for the carbon–zinc battery under different working conditions. It is obvious that PLM is the best under almost all working conditions owing to Peukert’s Law’s ability to better model the rate-capacity effect.

## 7. Conclusions

Accurate estimation of battery SOC is very important to realize dynamic power management of sensor nodes and effective utilization of battery capacity. Owing to the limited resources of sensor nodes, SOC estimation methods are required to be as simple as possible in both measurement and computation. This makes the fitting models based on terminal voltage and cumulative models based on current consumption widely used in WSNs. Unfortunately, accuracy of existing methods is not high, and their adaptability to different primary batteries and working conditions is relatively poor. With this in mind, this paper initially analyzes the estimation error of the cumulative models based on current consumption theoretically and then proposes a lightweight SOC estimation method that applies Peukert’s Law to estimate the effective capacity of the battery, which considers the rate-capacity effect of the battery more reasonably.

Applicability of Peukert’s Law for the commonly used primary batteries was verified initially. Experimental results show that Peukert’s Law is applicable for both the alkaline and carbon–zinc battery. Then, different discharge conditions, including constant current, constant resistance, and emulated duty–cycle discharge, were used to compare the proposed method with similar estimation methods. Experimental results show that the error variation in the cumulative models based on current consumption conforms to the theoretical analysis very well. Additionally, the proposed method is superior to existing methods for different batteries and working conditions, which benefits from Peukert’s Law’s ability to better model the rate-capacity effect of the battery.

However, it can also be seen that, similar to existing lightweight methods, accuracy of the proposed method also becomes obviously worse under duty–cycle discharge. This is because although Peukert’s Law can better model the rate-capacity effect, it cannot affect the recovery effect, another important nonlinear characteristic of the battery. Therefore, in the future, the impact of other nonlinear effects of the battery, such as the recovery effect, needs to be considered to further improve the accuracy of lightweight SOC estimation methods. Additionally, more comprehensive discharge tests will be conducted to study the performance of SOC estimation more deeply.

## Figures and Tables

**Figure 1 sensors-23-01036-f001:**
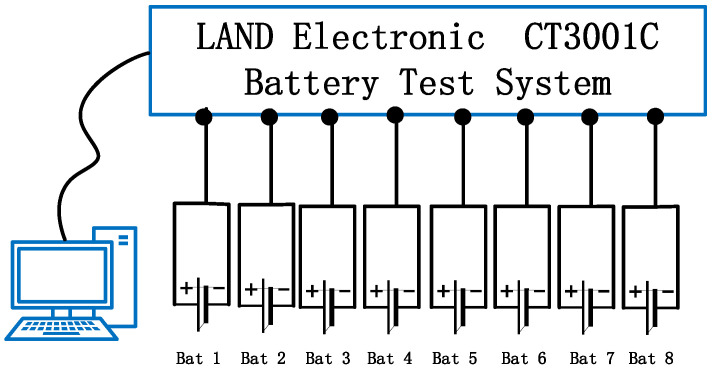
The microcurrent battery test system with eight independent programmable channels for different discharge conditions including constant current, constant resistance, and emulated duty-cycle discharge.

**Figure 2 sensors-23-01036-f002:**
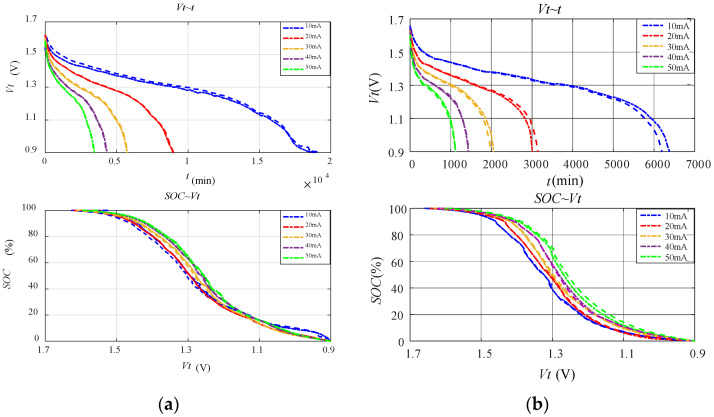
Discharge characteristics of alkaline and carbon–zinc batteries under constant current discharge tests: (**a**) alkaline battery; (**b**) carbon–zinc battery.

**Figure 3 sensors-23-01036-f003:**
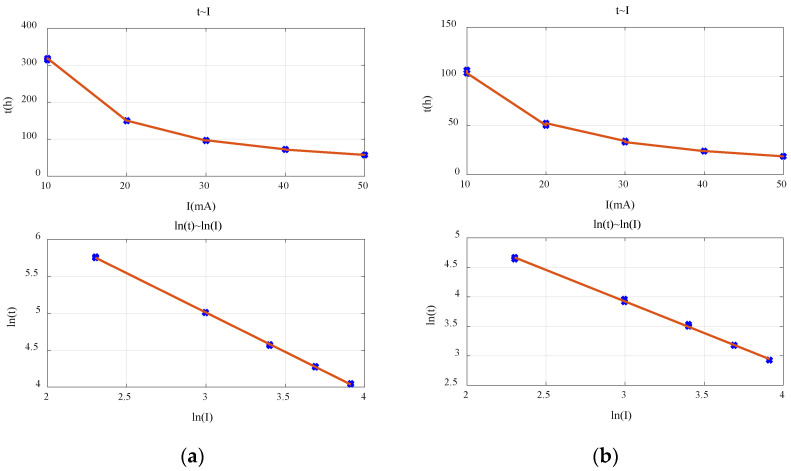
The relationship of discharge rate and discharge time: (**a**) alkaline battery; (**b**) carbon–zinc battery.

**Figure 4 sensors-23-01036-f004:**
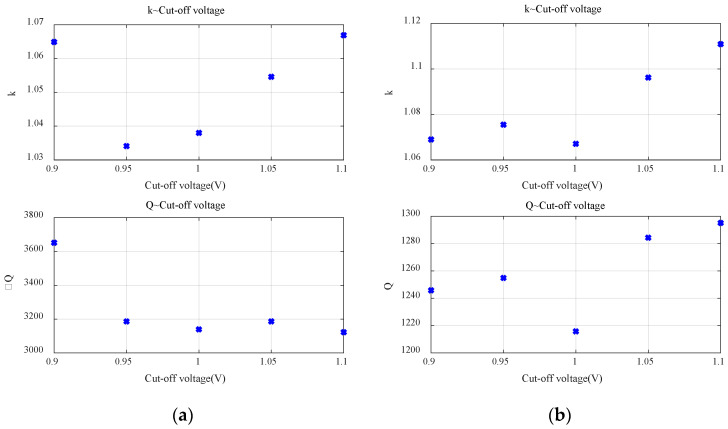
Effect of the cut-off voltage on *k* and *Q*: (**a**) alkaline battery; (**b**) carbon–zinc battery.

**Figure 5 sensors-23-01036-f005:**
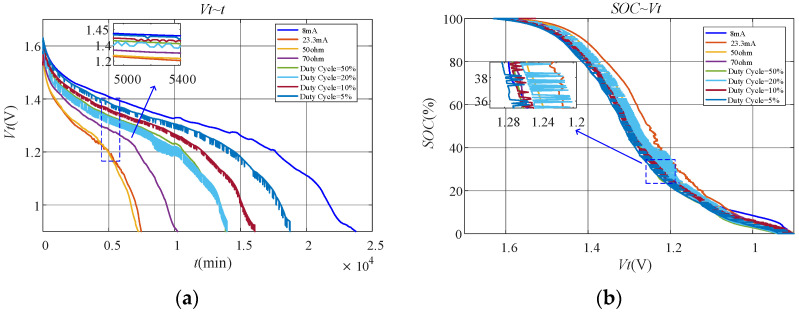
Discharge characteristics of the alkaline battery under different working conditions for verification: (**a**) Vt~t; (**b**) SOC~Vt.

**Figure 6 sensors-23-01036-f006:**
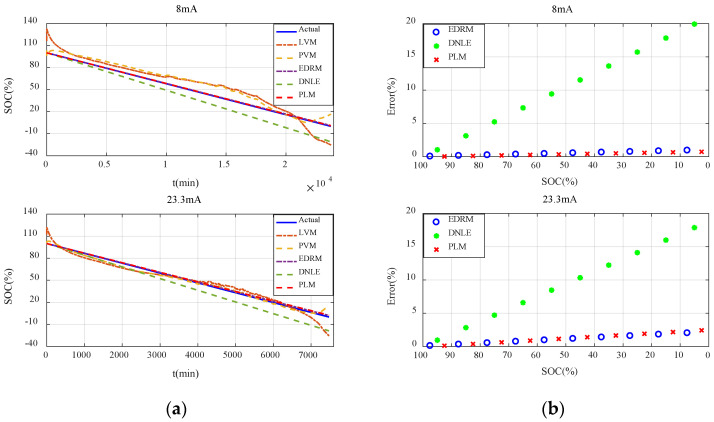
Estimation accuracy of SOC under constant current discharge for the alkaline battery: (**a**) actual SOC vs. estimated SOC; (**b**) average error for every 10% SOC interval.

**Figure 7 sensors-23-01036-f007:**
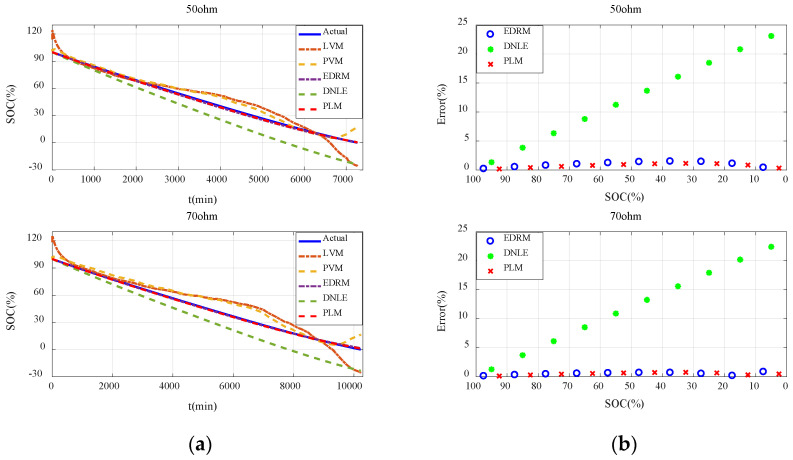
Estimation accuracy of SOC under constant resistance discharge for the alkaline battery: (**a**) actual SOC vs. estimated SOC; (**b**) average error for every 10% SOC interval.

**Figure 8 sensors-23-01036-f008:**
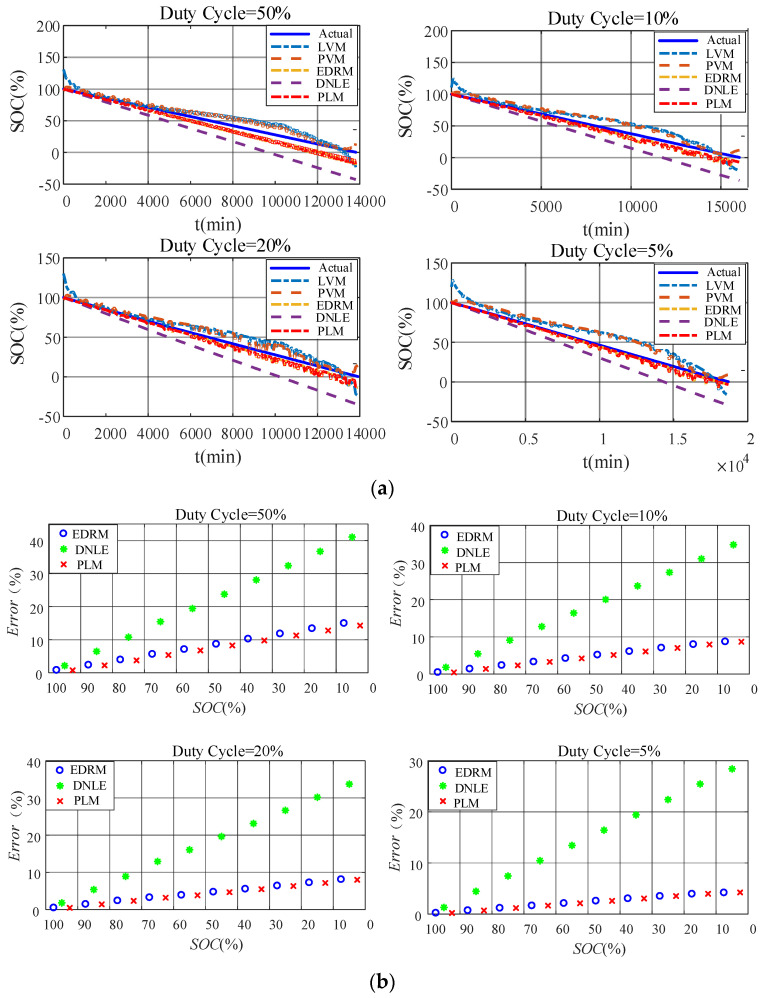
Estimation accuracy of SOC under duty−cycle discharge for the alkaline battery: (**a**) actual SOC vs. estimated SOC; (**b**) average error for every 10% SOC interval.

**Figure 9 sensors-23-01036-f009:**
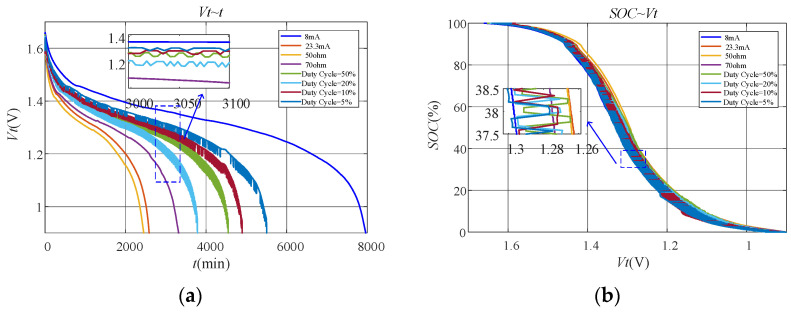
Discharge characteristics of the carbon–zinc battery under different working conditions for verification: (**a**) Vt~t; (**b**) SOC~Vt.

**Figure 10 sensors-23-01036-f010:**
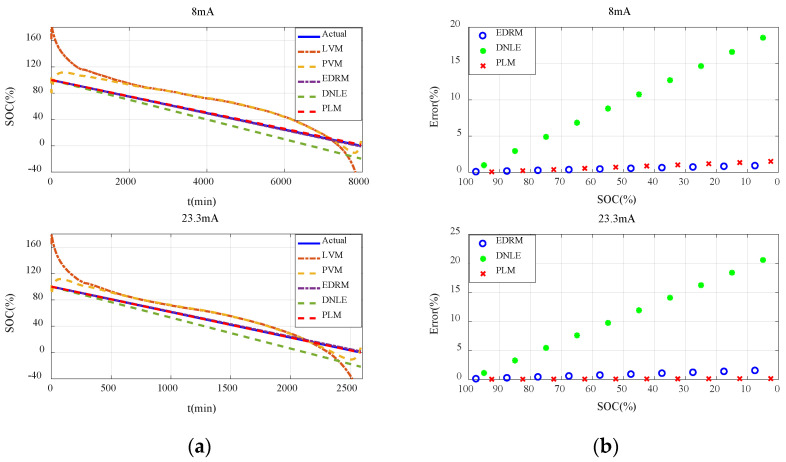
Estimation accuracy of SOC under constant current discharge for the carbon−zinc battery: (**a**) actual SOC vs. estimated SOC; (**b**) average error for every 10% SOC interval.

**Figure 11 sensors-23-01036-f011:**
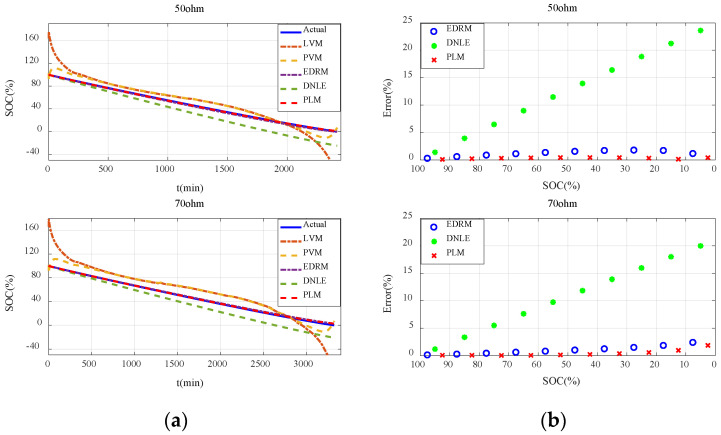
Estimation accuracy of SOC under constant resistance discharge for the carbon−zinc battery: (**a**) actual SOC vs. estimated SOC; (**b**) average error for every 10% SOC interval.

**Figure 12 sensors-23-01036-f012:**
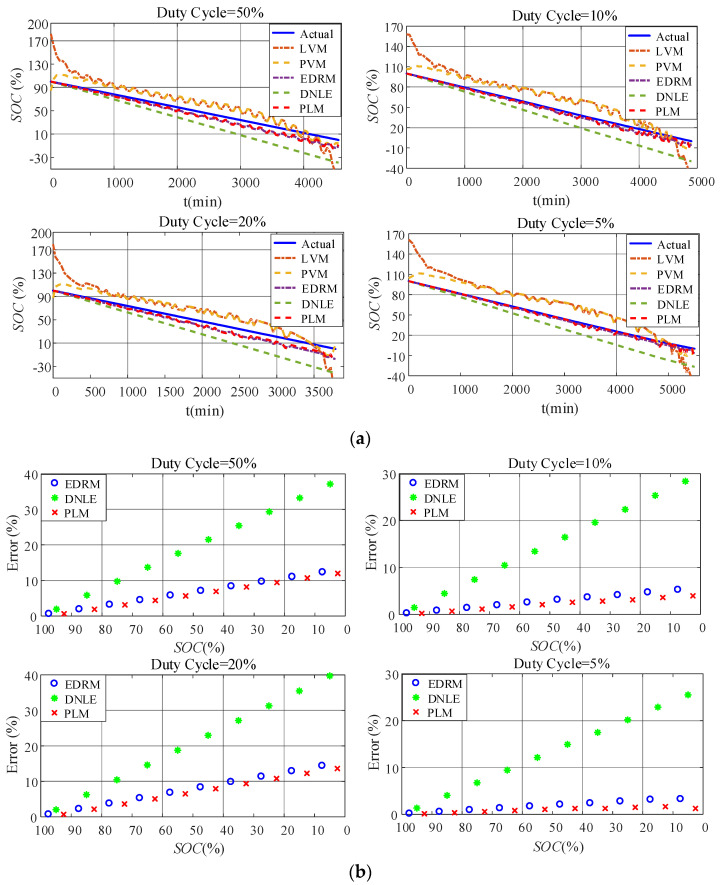
Estimation accuracy of SOC under duty−cycle discharge for the carbon−zinc battery: (**a**) actual SOC vs. estimated SOC; (**b**) average error for every 10% SOC interval.

**Table 1 sensors-23-01036-t001:** Variables used in subsequent analysis.

Variable	Description
*SOC_est_*	estimated SOC
*SOC_real_*	actual SOC
*V_t_*	terminal voltage
*I*	discharge current
Δ*t*	sampling interval for *V_t_* and *I*
*I_i_*	discharge current at the *i*th sampling interval
*a_i_*	coefficients of the polynomial function
*k*	Peukert constant
*Q*	empirical constant of Peukert’s Law
*C_est_(I)*	estimated available capacity under current *I*
*C_nominal_*	nominal capacity of battery
*C_real_(I)*	actual capacity under current *I*

**Table 2 sensors-23-01036-t002:** The fitted SOC estimation methods for the alkaline battery.

Fitted Methods	Alkaline Battery
LVM	$SOC=216.65V_{t}-220.38$
PVM	$SOC=-1212.53V_{t}^{3}+4627.91V_{t}^{2}-5618.21V_{t}+2208.14$
EDRM	$SOC=(1-\frac{\sum_{i=0}^{i=n}I_{i}\Delta t}{0.27\times {I_{n}}^{2}-23.56\times I_{n}+3366})\times 100\%$
DNLE	$SOC=(1-\frac{\sum_{i=0}^{i=n}{I_{i}}^{1.06}\Delta t}{2994.98})\times 100\%$
PLM	$SOC=(1-\frac{(\sum_{i=0}^{i=n}I_{i}\Delta t)\times I_{n}^{0.06}}{3651.89})\times 100\%$

**Table 3 sensors-23-01036-t003:** The fitted SOC estimation methods for the carbon–zinc battery.

Fitted Methods	Carbon–Zinc Battery
LVM	$SOC=338.35V_{t}-377.23$
PVM	$SOC=-1775.27V_{t}^{3}+6731.81V_{t}^{2}-8153.99V_{t}+3186.82$
EDRM	$SOC=(1-\frac{\sum_{i=0}^{i=n}I_{i}\Delta t}{-0.03\times {I_{n}}^{2}-1.13\times I_{n}+1063})\times 100\%$
DNLE	$SOC=(1-\frac{\sum_{i=0}^{i=n}{I_{i}}^{1.07}\Delta t}{1025.76})\times 100\%$
PLM	$SOC=(1-\frac{(\sum_{i=0}^{i=n}I_{i}\Delta t)\times I_{n}^{0.07}}{1245.84})\times 100\%$

**Table 4 sensors-23-01036-t004:** Average error of SOC estimation for the alkaline battery under different working conditions.

Working Conditions		PLM	EDRM	DNLE	PVM	LVM
constant current	8 mA	**0.39**	0.50	10.49	8.83	10.88
	23.3 mA	**1.07**	1.27	9.42	2.55	5.07
constant resistance	50 Ω	**0.73**	0.95	13.14	4.30	6.80
	70 Ω	**0.51**	0.52	12.77	7.18	8.91
emulated duty cycle	5%	**1.61**	2.35	15.30	9.92	11.68
	10%	**4.83**	4.84	18.90	8.62	10.36
	20%	**5.13**	5.18	21.46	6.32	8.50
	50%	**7.08**	9.04	24.78	8.92	8.83

**Table 5 sensors-23-01036-t005:** Average error of SOC estimation for the carbon–zinc battery.

Working Conditions		PLM	EDRM	DNLE	PVM	LVM
constant current	8 mA	0.79	**0.46**	9.77	17.34	22.52
	23.3 mA	**0.10**	0.79	10.85	10.14	15.02
constant resistance	50 Ω	**0.32**	1.19	13.30	8.54	13.33
	70 Ω	**0.48**	1.03	11.28	12.23	17.15
emulated duty cycle	5%	**1.08**	2.22	16.38	14.74	20.29
	10%	**2.36**	3.07	16.20	14.90	19.35
	20%	**7.34**	7.79	21.29	14.36	18.97
	50%	**6.34**	6.54	19.68	13.15	17.65

## Data Availability

Not applicable.

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
