# Peer review of "Peukert’s Law-Based State-of-Charge Estimation for Primary Battery Powered Sensor Nodes"

_sensors, 2023, doi:10.3390/s23021036_

Round 1
Reviewer 1 Report
1、The structure is too cumbersome. For example, in Sections 0, 1.1 and 1.2, the background introduction is too tedious and useless. Your work focuses on Peukert' law, so a brief introduction to the WSN is enough. The readers cannot get the points even after 3 pages have been read. The manuscript structure should be overall improved.
2、It just seems like a lab verification of the Peukert' law which has long been a well-acknowledged law. What's your innovative contributions?
3、How do you consider the fact that the smaller capacity by a bigger current rate is partly ascribed to the fixed cut-off terminal voltage in discharging process? For example, 2C discharge leads to a bigger ohmic polarization overpotential than 1C, so 2C discharge will reach the cut-off voltage earlier, which of course, to some extent, release less capacity.
4. How do you model the influence of temperature?
5. In the verifications, just a constant current was used in a discharging process. Then, how do you consider varying current rate in a discharging process? For example, if first-1C-then-2C, the denominator in (11) will be first-big-then-small. I suggest the following expression of (11) can be tried: 1-Σ(Iiδt/(Q/(Iik-1)). That is, the capacity consumptions between steps are independent, with a varying denominator.

Author Response
Dear Reviewer:
I have replied to your comments point-by-point and upload it as a Word file,please check it, thank you!

Reviewer 2 Report
The presentation style of the manuscript should be revised. Therefore, the paper should be revised by considering the following issues:
MAJOR ISSUES
+ Introduction section should be revised considerably.
+ The main contributions of the paper should be clearly given as a separate subsection in the introduction section.
+ The organization of the paper should be clearly given as a separate subsection in the introduction section.
+ The related work and so bibliography should be improved by adding more references.
+ Most of the references in this paper are mostly recent publications (within the last 5 years) and relevant. On the other hand, the bibliography should be improved by adding most recent references.
+ Preamble information is required between section"1. Related Works" and subsubsection "1.1. Power Supply for Sensor Node".
+ “Problem Definition and System Model” should be given more clearly as a separate section.
+ In Equation (3), (4) and (5), proper fraction should be used instead of "/".
+ Equation (6) should be written as "Q=t*I^k". (Left side should be on right side). Similarly, Equation (9) should be written as "Q=C_{est}*I^{k-1}". An equation should be written as in the form of "y=f(x)".
+ The proposed scheme performs well. The motivation behind it should be explained better.
+ The figures/schemes are generally clear. They show the data properly. It is not difficult to interpret and understand them. On the other hand, Figure 1 should be explained better by adding more information to its caption.
+ Section 4. Experimental Evaluation should be definitely improved. Many more figures should be given in the numerical results section. Figures should be clearly explained, especially in the text/main body of the paper.
+ Preamble information is required between section "4. Experimental Setup" and subsection "4.1. Testbed for Discharging".
+ The conclusion should be improved by giving the key results and main contributions more clearly.
+ Future work part should be given in the conclusion section.
MINOR ISSUES
+The grammatical errors and typos should be fixed.
+The authors should adjust the (section) counter as "0" instead of the default value "-1".
+Size of Figure 3, 4, 5, 6, 7, 8, 9, 10 and Table 2 should be reduced to be kept in page margins.
+The references in the bibliography should be given in the same style. The following link should be checked: https://www.mdpi.com/authors/references
Author Response

(The authors gave the same response as above.)

Round 2
Reviewer 1 Report
Most concerns have been responded. Except, the writing of the manuscript should be further improved to get rid of grammatical errors. Most concerns have been responded.
Reviewer 2 Report
The paper is acceptable in its current form.